# Effectiveness of App-Based Intervention to Improve Health Status of Sedentary Middle-Aged Males and Females

**DOI:** 10.3390/ijerph19105857

**Published:** 2022-05-11

**Authors:** María Martínez-Olcina, Bernardo José Cuestas-Calero, Laura Miralles-Amorós, Manuel Vicente-Martínez, Javier Sánchez-Sánchez

**Affiliations:** 1Department of Analytical Chemistry, Nutrition and Food Science, Faculty of Sciences, University of Alicante, 03690 Alicante, Spain; laura.miralles@ua.es; 2Faculty of Sports, Catholic University of Murcia (UCAM), 30107 Murcia, Spain; bjcuestas@alu.ucam.edu; 3Faculty of Health Science, Miguel de Cervantes European University, 47012 Valladolid, Spain; mvmartinez11006@alumnos.uemc.es; 4School of Sport and Science, European University of Madrid, 28670 Madrid, Spain; javier.sanchez2@universidadeuropea.es; 5IGOID Research Group, University of Castilla de la Mancha, 13071 Ciudad Real, Spain

**Keywords:** health, body composition, bioelectrical impedance analysis, personalized nutrition, mobile app, weight management, blood pressure, mHealth

## Abstract

Background: Adherence to a nutritional program and physical activity are the fundamental aspects of treatment for weight loss and associated problems. Previous research has shown that self-monitoring using a mobile device improves self-management. Methods: A total of 35 subjects (40.6 ± 9.24 years) participated in the study. During the control period (3 months), they received physical exercise guidelines and a personalized nutritional program, with the aim of promoting health status. In the experimental period (3 months), there was also a connection between the physical world (health care processes) and the digital world (app). All participants had their body composition and cardiovascular variables measured. They also underwent calcaneal densitometry to determine bone quality. Descriptive statistics, correlations and analysis of variance were performed (by a researcher who was not involved in the data collection) to study the changes between before and after interventions, as well as to make a comparison between treatments. Results: The use of an app, in which there exist a prediction of the evolution, messages of results and advice, among others, mediated by the assistance of dietitians/nutritionists and sports scientists, had a positive impact on the improvement of health parameters, showing significant differences in all variables except troponin. Conclusions: The combination of healthy habits with the use of the app provided benefits, improving health.

## 1. Introduction

Obesity is defined by WHO [1] as abnormal or excessive fat accumulation that can impair health. It is a growing threat to public health and well-being [2] and has become an epidemic in both developed and developing countries. Obesity and cardiometabolic dysfunction are diseases with very complex management due to their multifactorial nature involving environmental, genetic and psychosocial factors [3]. Obesity is one of the main risk factors for the development of type 2 diabetes, hypertension, coronary heart disease and stroke, as well as certain types of cancer: endometrial, breast and colon [4]. 

An adequate diet and physical activity are key treatments for the prevention of these diseases. Scientific evidence has shown that the adoption of the Mediterranean diet is a protective factor against the appearance of several types of cancer, cardiovascular diseases, aging and obesity [5] due to the amount of nutrients with anticancer, anti-inflammatory and anti-obesity properties that jointly contribute to the maintenance of health status. The antitumor effects of the Mediterranean diet are mainly due to the combination of antioxidant elements, fiber and polyunsaturated fats. Therefore, this dietary pattern is essential as a preventive measure against the onset of cancer and other chronic diseases, as well as to reduce health care costs.

Monitoring physical activity should be considered as a central factor when approaching precision nutrition [6]. Performing physical activity and exercise, key components of energy expenditure and balance, also lead to structural changes in muscles, an increase in the number of mitochondria in fiber, the secretion of metabolically beneficial hormones with the reversal of muscle insulin resistance and the reduction of hepatic lipogenesis [7]. 

The lack of awareness of these diseases on the part of the population is due, among other aspects, to the impossibility of patients to visualize the complications and adverse effects that metabolic pathologies have on health [8,9,10]. In addition, the alterations that these parameters cause to the physiological state of the organism are not visible to the naked eye, but are only reflected in tests, and, therefore, on many occasions, they are not recognized as problematic or are simply overlooked in the people who present them.

There are numerous mobile apps available for download that target health and fitness [11]. Their popularity and potential to influence health-related behaviors make their integration into medical practice imminent. It has been seen that, with the use of mHealth applications, it is possible to increase the uptake of health prescriptions [12], encouraging healthy behaviors related to, for example, dietary habits, weight management, physical activity, addictive behaviors (e.g., smoking) and mental health (e.g., stress and depression management), among others [13,14]. They also allow for continuous monitoring, which provides the basis for individualized feedback and goal setting.

Therefore, in the present investigation, the development of an app is intended—through self-management and visualization of the altered parameters and the way in which they will evolve in the body in the short, medium and long term—to increase the population’s commitment to these pathologies in order to improve their health [15,16,17,18,19,20]. 

The initial hypothesis is that, during the period of use of the app, the improvement will be greater than when the app is not used. Therefore, the objective is to compare the improvement in certain health parameters (body composition, cardiovascular parameters and bone mineral density) after an intervention based on a personalized nutritional program and physical exercise with and without the use of a mobile app. As such, the aim is to improve the value chain in the health sector dedicated to the prevention and treatment of pathologies related to nutrition and physical exercise.

## 2. Materials and Methods

### 2.1. Study Design

A clinical trial study was carried out to determine the efficacy of a mobile application, personalized nutritional program, and physical exercise to improve different health parameters. The follow-up of the study is shown in Figure 1.

### 2.2. Participants

A total of 35 participants aged 40.6 ± 9.24 years recruited at Nutrición Selecta, Elche, during 2020–2021 participated in the study. Of the total sample, 16 were men and 19 were women. All subjects participated both during the control period (nutritional intervention + exercise), while the app was being developed, and during the experimental period, once the app was ready for use, the same subjects received the nutritional intervention + exercise + app. This allowed for a comparison to be made between a conventional follow-up of diet and physical exercise and a follow-up using the app. As inclusion criteria, they had to be over 21 years of age; be slightly overweight, overweight, or obese and/or have slightly elevated blood pressure; have access to a smartphone; be able to attend all the proposed visits; and give written consent prior to participation. Exclusion criteria included lack of time to attend the corresponding follow-ups; physical difficulties to follow exercise guidelines; chronic nutritional disease; intolerances, such as Crohn’s disease, celiac disease, or diabetes; and people who were already following the guidelines of external nutritionists on their own.

### 2.3. Declarations: Ethical Approval, Consent to Participate and Consent to Publish

The present study was conducted in accordance with the standards of the Helsinki declaration. The Human Research Ethics Committee of the University of Alicante (Spain) granted approval to conduct the trial (UA-2021-03-27), and all study participants gave written consent prior to participation. In addition, the research was registered in the official clinical trials database, ClinicalTrials.gov, obtaining the registration number NCT05093803. In addition, the investigators maintained the confidentiality of all personal data of the participants, coding personal information for this purpose.

### 2.4. Intervention

The intervention lasted 6 months (3 for control intervention and 3 for app intervention). During the control period, the subjects received physical exercise guidelines and an adapted, personalized nutritional program, with the aim of promoting health status. During the following 3 months, the use of the mobile app was added. For the design of the plans, an internal database was used, based on the nutritional calibration system by exchanges. All volunteers were prescribed a hypocaloric diet with 1.4 g/kg body weight/day of protein, with the aim of avoiding a loss of muscle mass, based on the principles of the Mediterranean diet [21]. The physical exercise recommendations were based on the guidelines proposed by the American College of Sports Medicine (ACSM) [22].

In the experimental period, there was also a connection between the physical world (health care processes) and the digital world (app). As such, the volunteers (with chronic cardiovascular pathologies associated with body composition, such as obesity and hypertension) had a visual predictive representation in which self-care and a personalized healthy lifestyle were promoted. Regarding the app, the health coach was responsible for activating each individual’s personal user account at the initial meeting. Based on the results of the health profile, a nutritional program was designed, in addition to physical exercise recommendations. Through the app (Figure 2), subjects could view their results graphically, as well as identify how they are doing based on the reference. In addition, depending on the results obtained, the app launched different result messages and advice regarding diet, physical activity, and general health. Table 1 shows the English translation of the information presented in the images.

The application displayed clear one-way messages with specific goals, challenges for healthy living, information or questions, according to the values obtained by each of the participants (see examples in Table 2). Currently, messages are only sent in Spanish. An independent dietitian and a sport scientist reviewed the content and wording of the messages.

### 2.5. Study Variables

Figure 3 shows the development of each of the measurements taken from the subjects.

#### 2.5.1. Body Composition

In the present investigation, weight, and body composition (fat mass, body water and visceral fat) were assessed using Tanita BC-545N (Tokyo, Japan). Subjects were barefoot, without socks or stockings, with feet fully supported and placed in the electrode area. They had to remove all metallic elements (bracelets and rings). The measurements were taken at the same time of day to avoid hydration bias, always allowing 3 h to elapse from the time of the meal and/or training.

The height of the subjects was measured with a SECA 123 stadiometer (Hamburg, Germany). From the body mass and height data, BMI (kg/m^2^) was calculated. Waist (cm) and hip (cm) perimeters were also measured, from which the waist-to-hip index (WHI) was obtained. The WHI is a specific anthropometric measure of intra-abdominal fat levels. WHO establishes normal WHI levels of 0.78–0.94 in men and 0.71–0.84 in women; higher values indicate abdominal visceral obesity, which is associated with an increased cardiovascular risk and an increased probability of contracting diseases such as diabetes mellitus and arterial hypertension.

#### 2.5.2. Cardiovascular Parameters

Blood pressure (systolic and diastolic blood pressure) was measured using an Omron M4 Intelli IT (WendenstraBe, Hamburg, Germany) sphygmomanometer. Patients were seated and relaxed. Blood parameters cholesterol, triglycerides, troponin I (cTnI) and troponin T (cTnT) were measured using blood analysis. This sample collection and analysis were performed by an external laboratory. Resting heart rate and maximum heart rate measurements were monitored with an electrocardiographic stress test. For this purpose, the Bruce test was performed [23,24], in which relative required power outputs or work rates progressively increase in 3 min stages. cTnI and cTnT are highly specific “cardiac troponin” proteins (cTns) involved in myocardial cell damage and are key factors in the diagnosis of acute coronary syndromes and myocardial necrosis.

#### 2.5.3. Bone Mineral Density

Bone quality was measured using ultrasound densitometry (US), performed on the heel (Achilles EXP II, GE Healthcare, Chicago, IL, USA). US does not measure bone mass. The parameters measured by US are attenuation or BUA (broadband ultrasound attenuation (BUA) and the speed of sound through bone or speed of sound (SOS). As a combination of these, the Stiffness index was obtained. This was calculated using a formula previously used in other studies [25]. Additionally, to provide quantitative data, this index evaluates qualitative aspects, such as the elasticity, structure and geometry, of the bone. It is the only densitometer that analyzes these aspects of microarchitecture, which are increasingly relevant as risk factors for fracture [26,27].

### 2.6. Statistical Analysis

Statistical analysis of the data was performed with the JAMOVI statistical program (Sydney, Australia) by a researcher not involved in the data collection. For descriptive statistics (mean ± standard deviation) and inferential analysis, the Shapiro–Wilk test was used to establish the normality distribution. The analysis of differences between initial and final values was performed using the t-test for related samples. Statistical differences between the different times and periods were tested using analysis of variance (ANOVA) with Bonferroni post hoc comparisons. The significance level was set at *p* < 0.05. Effect sizes were calculated with the η^2^ statistic; the effect of η^2^ ≥ 0.01 indicates a small effect, ≥0.059 a medium effect and ≥ 0.138 a large effect.

## 3. Results

### 3.1. Socio-Demographic Data

Of the total sample, 16 were men and 19 were women (Table 3). All participants were involved in both the control period and experimental period (app).

### 3.2. Body Composition

Regarding body composition variables, significant differences were observed in all the variables studied, namely, weight, BMI, the percentage of fat mass, body water and visceral fat.

For the weight variable, significant decreases were observed between the initial app vs. final app (*p* < 0.001), initial control vs. final app (*p* = 0.007) and initial control vs. final control (*p* = 0.001). Therefore, for BMI, differences were found between the same moments, that is, initial app vs. final app (*p* < 0.001), initial control moment vs. final app (*p* = 0.002), initial control moment vs. final control moment (*p* = 0.002), and between the two final moments (*p* = 0.004).

Regarding the variable’s percentage (%) of fat and body water, both over time and between periods, significant differences were observed (Table 3). After performing the post hoc analysis, in the fat % variable, differences were observed between the initial and final time in both the app period (*p* < 0.001) and the control period (*p* = 0.026). However, for the visceral fat variable, there were only significant differences in terms of time, that is, initial vs. final control period (*p* = 0.006) and initial vs. final app period (*p* = 0.006), and not between periods (Table 4). In all the variables, the results improved after the intervention; therefore, this is translated into a lower % fat, higher body water and lower visceral fat. It should be noted that, in the body water variable, in addition to the differences between the initial and final time of both the control period (*p* = 0.003) and app (*p* < 0.001), there is a significant difference at the final time between the two periods (*p* = 0.004).

Regarding the perimeters, differences were observed between the initial values and the final assessments in the app period, with a significant decrease in both waist circumference (*p* < 0.001) and hip circumference (*p* < 0.001). In addition, significant differences were observed between the initial results (final results of the control period) and the final results (final results of the app period), both in waist circumference (*p* < 0.001) and hip circumference (*p* = 0.002).

### 3.3. Cardiovascular Parameters

In the analysis of the circulating parameters of the study subjects, focusing on the lipid profile (Table 5), a significant decrease was observed in the variable triglycerides (mg/dL) between the initial and final assessments of the app period (*p* = 0.031). As for cholesterol (mg/dL), significant improvements were observed in both the control period (*p* = 0.004) and the app period (*p* < 0.001), in addition to significant differences (*p* < 0.001) between the final assessments of both periods.

Regarding blood pressure analysis, there was a significant decrease in both periods. For systolic blood pressure (SBP), significant differences were observed between the initial and final measurements of the app period (*p* < 0.001); between the initial time of the control period and the final time of the app period, that is, between the first and last measurements of the total investigation (*p* = 0.004); and, finally, between the two final times (*p* < 0.001). As can be seen in Table 4, the ASR decreased according to time. For diastolic blood pressure (DBT), significant differences were also observed between the initial and final measurements of the app period (*p* < 0.001), between the initial time of the control period and the end of the app period (*p* = 0.010) and between the two final times (*p* = 0.008).

Moreover, with regard to the resting heart rate measured with an electrocardiogram and the maximum heart rate obtained after the stress test, shown in Table 4, a significant decrease was observed in both the resting heart rate and the maximum heart rate between the initial and final moments of the app period (*p* < 0.001 and *p* < 0.001 for resting heart rate and maximum heart rate, respectively), between the initial control and final app moments (*p* = 0.026 and *p* < 0.001) and between the final control and final app moments (*p* = 0.033 and *p* < 0.001).

After the intervention, a slight decrease was observed in cTnI and cTnT (Figure 4); however, these changes were not significant.

### 3.4. Bone Mineral Density

Regarding bone quality (Table 6), significant differences were observed in the BUA and Stiffness variable in the control period between the initial and final time (*p* = 0.015 and *p* = 0.024, respectively) and in the SOS variable between the initial time of the control period and the end of the app period (*p* = 0.027), that is, between the first and last evaluation of the total project.

## 4. Discussion

Cardiovascular diseases include a broad spectrum of disorders affecting the heart and blood vessels, and they are the leading cause of death in the world [4,21]. Health professionals should encourage measures that impact lifestyle, but their adoption depends on the patient’s understanding of his or her own problems, motivations and conditions associated with the treatment.

Adequate nutrition and physical activity are key to the prevention of these diseases. Following a dietary pattern based on the Mediterranean diet [7] has beneficial effects. This dietary pattern is characterized by a balanced intake of fruits and vegetables, fish, cereals and polyunsaturated fats, with a reduced consumption of meat and dairy products. The value of this diet lies in its ability to preserve health status and improve longevity [5]. 

The lack of awareness of these diseases by the population is due, among other aspects, to the impossibility of visualizing the complications and adverse effects of metabolic pathologies on the patient’s own health [8,9,10,28]. In addition, the alterations that these parameters cause to the physiological state of the organism are not visible to the naked eye but are only reflected in the tests, and, therefore, on many occasions, they are not recognized as problematic or are simply overlooked in the people who present them. 

In this context, the present research aimed to provide a solution to the needs, improving the health care value chain dedicated to the prevention and treatment of pathologies related to nutrition and physical exercise.

After carrying out different interventions, namely, the nutritional program + physical exercise recommendations (CTRL) and nutritional program + physical exercise recommendations + app (app), significant improvements were observed in most of the variables studied. As far as body composition (BC) is concerned, it was observed that patients had significant improvements after the first period (CTRL) but also in the second period (app), and, additionally, for the variables BMI, body water, waist circumference and hip circumference, there were differences between the final moments of the CTRL and app periods. It seems, therefore, that an intervention based on a personalized nutritional program together with physical activity recommendations improves all BC variables; however, when an app is used, the difference is statistically greater, i.e., there is a greater range of improvement. This was to be expected, since, in some studies [6,8,9,10,20,28], it has been observed that, through the implementation of these new technologies, patients with chronic diseases achieved an improvement in their state of health.

As for the lipid variables, only the app period (app + nutritional program + physical exercise) significantly improved triglyceride blood values; however, both the CTRL and app periods improved cholesterol. Nevertheless, there were differences between the measurements of the two post intervention periods, with the values of the app period being lower, i.e., closer to the values considered healthy. These results coincide with those of the research carried out by Raquel Debon et al. [29], in which they observed that cholesterol showed significant improvements unlike triglycerides, which require a longer period of dietary re-education and healthy habits to achieve more significant changes.

The improvement in lipid profile is a remarkable result, considering that more than half of the decrease in cardiovascular mortality has been attributed to changes in population risk factors, especially lower levels of cholesterol and blood pressure, among others [30]. 

The same occurs in the case of systolic blood pressure, diastolic blood pressure, resting HR and maximum HR, therefore, once again confirming the positive effect of the app in the improvement of all variables. This suggests that, if the subjects had continued with the intervention for longer, superior improvements would have been achieved. For bone quality, it does not appear that the app presents statistically different improvements compared to the nutritional program + exercise intervention. The same occurs with cTnI and cTnT. 

In view of the results, it can be affirmed that the use of an app, in which there exist a prediction of evolution, result messages and advice, among others, mediated by the assistance of dietitians/nutritionists and sport scientists has a positive impact on the improvement of health parameters. Systematic reviews on the efficacy of health apps are sometimes contradictory. Madison Milne-Ives et al. [31] noted, after examining 52 randomized controlled trials evaluating the effectiveness of mobile health apps, that there was little meaningful evidence to support the effectiveness of mobile health apps. However, both Han and Lee [32,33] and Payne et al. [13] reported that most of the apps reviewed were effective in improving health-related behaviors. 

During the 3 months in which the use of the app was added to the nutritional program and physical exercise was performed, it was observed that the improvements were much greater when the app was used than when the same plan was followed without using it. Healthy habits, such as following a nutritional program and performing physical exercise regularly, had obvious beneficial effects on the health conditions of the participants, but combining these elements with the use of the app boosted the benefits, improving their health. These results suggest that, if the subjects had continued with the intervention for a longer time, the improvements would have been even greater.

The strengths of the present study are that it was a controlled trial and, although sample randomization could not be performed, it was possible to compare the two types of intervention in the same population sample. The present study provides an ideal environment and infrastructure to support the promotion of health to a large audience. The results were objectively measured (anthropometric measurements, blood pressure and laboratory values). In addition, the Nutrition Selecta app is available for Android and iOS devices.

As a main limitation, it is important to emphasize that, in addition to the means offered to the patients, the willingness of the patients to follow the recommendations is also important in this type of intervention; there were participants who had reductions close to what was expected, others even more, but also others whose improvement was half of what was expected. Not all individuals required the same time as others to carry out these adaptations. The same individuals from the same company performed the measurements; however, the researcher who analyzed the results was not involved in the measurements.

Moreover, the same population sample took part in both periods (with and without app). In addition, the data were collected manually by the health personnel; however, in future studies, it is intended to use wearables, as these can act as an extension of the user’s body and mind, automatically providing information on bio-parameters, without the intervention of a health professional. At present, applications are not yet optimized for this type of device. However, this research shows that there is greater variability in the adaptations in bio-parameters using a health app. As such, it also helps health professionals to adapt the proposed strategies.

The use of the “Nutrition Selecta” app is a novel approach to delivering personalized, quality-tested, health interventions in a variety of lifestyle areas, using a combination of an app and conventional intervention strategies. The system is designed to be flexible and can be easily adapted based on outcomes. Future research should focus on performing this type of intervention with a larger sample of subjects and for a longer period, as well as measuring other parameters that may influence weight and fat loss, such as a lack of rest and chronic stress. In addition, the regulation and certification of the mobile app are also contemplated. It would be of great interest to add a control group, without nutritional planning, physical activity or mobile app use. 

## 5. Conclusions

The study shows evidence that the use of an application can have beneficial effects on the health conditions of patients; therefore, the use of technologies combined with health information is a positive development that can contribute to the therapeutic scheme of patients with overweight and arterial hypertension, among others, providing greater adherence to treatment, healthier habits and better health conditions.

## Figures and Tables

**Figure 1 ijerph-19-05857-f001:**
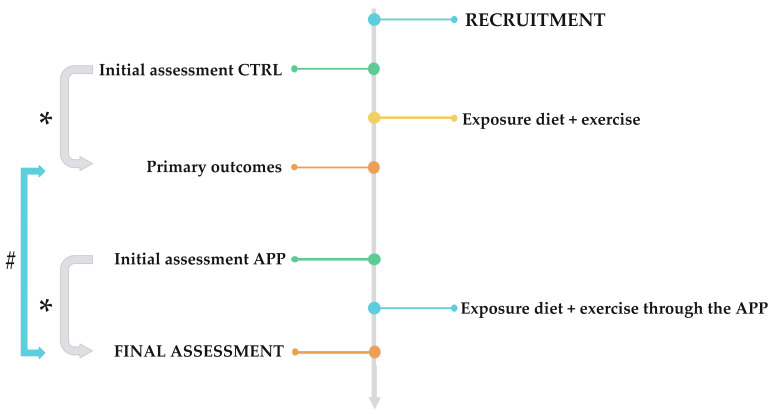
Clinical trial design. * = study if there are intra-period differences; # = study if there are inter-period differences. CTRL = control. app = mobile application.

**Figure 2 ijerph-19-05857-f002:**
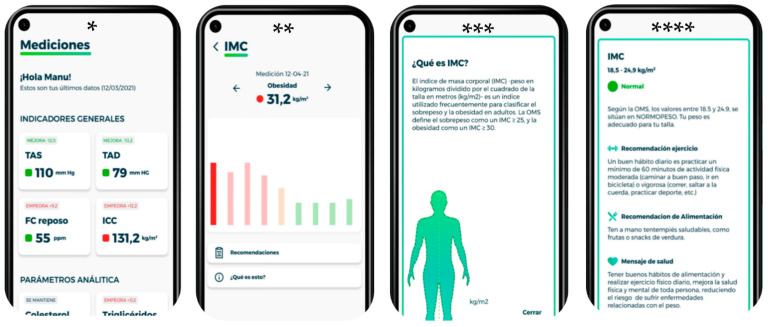
Actual representation of the mobile app used.

**Figure 3 ijerph-19-05857-f003:**
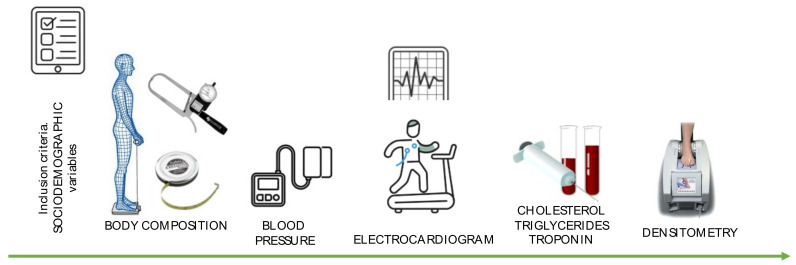
Summary of the assessment of health parameters included in the project.

**Figure 4 ijerph-19-05857-f004:**
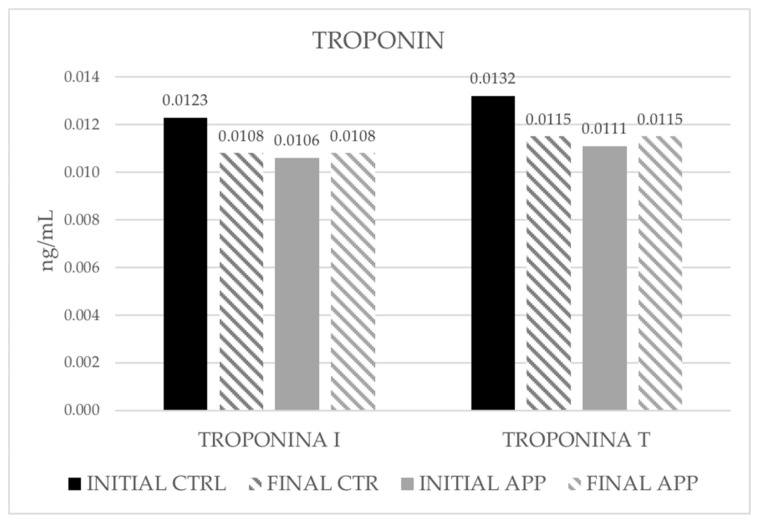
Descriptive (mean) of the variables troponin I and troponin T.

**Table 1 ijerph-19-05857-t001:** Information contained in the images of Figure 2 in English.

*	Measurements. General indicators. The image shows the variables: systolic blood pressure, diastolic blood pressure, resting heart rate and waist/hip ratio.
**	Once you click on one of the variables, the record of the different measurements appears (bar graph). Within the screen there is a section of recommendations and what is this?
***	What is BMI? Body mass index (BMI)—weight in kilograms divided by the square of height in meters (kg/m^2^)—is an index frequently used to classify overweight and obesity in adults. WHO defines overweight as a BMI > 25 and obesity as a BMI > 30.
****	BMI According to WHO, values between 18.5 and 24.9 are NORMAL WEIGHT. Your weight is appropriate for your size. Exercise recommendation: A good daily habit is to engage in a minimum of 60 min of moderate (brisk walking, cycling) or vigorous (running, jumping rope, sports, etc.) physical activity. Food recommendation: Have healthy snacks on hand, such as fruits or vegetable snacks. Health message: good eating habits and daily physical exercise help to improve the physical and mental health of everyone, reducing the risk of weight-related diseases.

**Table 2 ijerph-19-05857-t002:** Examples of application messages.

Variable	Result Message	Exercise Message	Eating Message	Health Message
BMI (kg/m^2^) ≥ 25.0	Overweight is caused by an abnormal or excessive accumulation of fat that can be detrimental to health.	Exercise helps regulate metabolism, causing an increase in metabolism by using energy reserves (glycogen and fat) to run the muscles.	A proper diet would improve your results, this is composed of a varied diet, with plenty of fruits and vegetables, avoiding processed products and alcohol.	Maintaining a healthy weight does not mean dieting. It is a lifestyle. There are simple steps you can take every day to keep your weight at healthy levels and reduce your risk of weight-related diseases and health problems.
Cholesterol ≥ 240 mg/dL	At this cholesterol level, the probability of suffering heart disease is twice as high as with values < 200 mg/dL.	Regular physical activity can help you control your weight and thus lower your cholesterol.	Eating foods rich in Omega 3 helps regulate total blood cholesterol. Among these foods are nuts, avocados, and oily fish.	High cholesterol affects the heart and blood vessels and increases the risk of developing cardiovascular disease.
Hight fat mass (%)	Your percentage of fat mass is HIGH. This increases the risk of heart disease and stroke.	Daily physical exercise has a beneficial effect on body composition. It helps to reduce skin folds, as well as the body fat index.	If you take in more calories than you burn, the excess calories are stored in your body in the form of fat cells. When the stored fat is not subsequently converted into energy, excess body fat is produced.	Excess body fat increases the risk of depression. Scientists at the University of Aarhus in Denmark conclude that excess body fat increases the chance of depression by up to 15%.
SBP 120–130 mmHg	Your systolic blood pressure is at HIGH values. Control these values regularly, having high blood pressure can have serious repercussions on your health.	Daily physical exercise (walking, running, cycling, swimming, etc.) for 30–60 min, 3 to 5 days a week, will help regulate your blood pressure.	Refined carbohydrates, especially sugar, can increase blood pressure. Some studies have shown that low-carbohydrate diets may help lower your levels.	There is conflicting research on smoking and high blood pressure, but what is clear is that both increase the risk of heart disease.

SBP = systolic blood pressure; kg = kilogram; m = meter; % = percentage; mg = milligrams; dL = deciliters; mmHg = millimeters of mercury.

**Table 3 ijerph-19-05857-t003:** Descriptive statistics (mean ± SD) of the sample separated by sex.

	Men		Women	
	Mean	SD	Mean	SD
Age	35.38	7.25	45.00	8.68
Height	176.05	7.62	162.77	5.42

SD = standard deviation.

**Table 4 ijerph-19-05857-t004:** Descriptive data (mean ± standard deviation) of body composition variables. Effect of the intervention (ANOVA) on blood pressure variables.

	CTRL				app									
	Initial		Final		Initial		Final		Effect Time			Effect Time × Period		
	Mean	SD	Mean	SD	Mean	SD	Mean	SD	F	*p*	η^2^_p_	F	*p*	η^2^_p_
Weight (kg)	83.00	16.4	81.9	16.2	78.5	15.5	72.6	15.0	206.7	< 0.001	0.752	93.4	< 0.001	0.579
BMI (kg/m^2^)	29.00	4.88	28.6	4.72	27.5	4.58	25.4	4.44	202.9	<0 .001	0.749	92.7	< 0.001	0.577
Fat mass (%)	30.2	10.3	29.5	10.8	28.1	10.4	26.1	9.73	55.5	<0 .001	0.450	11.4	0.001	0.143
Water (%)	50.6	7.05	51.4	7.57	53.7	8.07	57.6	8.84	120.6	<0 .001	0.639	55.5	< 0.001	0.449
Visceral fat	8.14	3.88	7.81	3.64	7.46	3.52	7.08	3.42	26.494	< 0.001	0.280	0.108	0.743	0.002
Hip (cm)	90.5	11.4	90.2	10.9	86.1	11.1	79.9	10.3	50.5	< 0.001	0.426	40.3	< 0.001	0.372
Waist (cm)	106.0	11.3	105.0	9.98	103.0	9.83	96.2	10.1	54.4	< 0.001	0.444	37.7	< 0.001	0.357
WHI	0.858	0.0974	0.858	0.0945	0.841	0.0918	0.833	0.0923	1.73	0.193	0.025	2.14	0.148	0.031

CTRL = control; SD = standard deviation; cm = centimeters; kg/m^2^ = kilograms/meter^2^; % = percentage; cm = centimeters; WHI = waist/hip index; F = F statistic; η^2^p = partial eta squared effect sizes. In the model, all differences were significant when *p* < 0.005.

**Table 5 ijerph-19-05857-t005:** Descriptive data (mean ± standard deviation) of cardiovascular parameters. Effect of the intervention (ANOVA) on blood pressure variables.

	CTRL				app									
	Initial		Final		Initial		Final		Effect Time			Effect Time x Period		
	Mean	SD	Mean	SD	Mean	SD	Mean	SD	F	*p*	η^2^_p_	F	*p*	η^2^_p_
CHO (mg/dL)	188.0	45.4	175.0	38.0	167.0	37.6	155.0	34.9	9.1	0.003	0.119	0.9	0.339	0.013
TG (mg/dL)	104.0	127.0	100.0	123.0	96.8	123.0	89.7	113.0	356.4	<0 .001	0.344	0.1	0.799	0.001
SBP (mmHg)	130.0	19.8	130.0	17.1	124.0	17.2	115.0	15.7	27.1	< 0.001	0.285	28.9	< 0.001	0.298
DBP (mmHg)	83.6	13.2	83.6	12.6	79.8	12.6	74.0	11.5	17.0	< 0.001	0.200	16.4	<0 .001	0.194
HR at rest	69.6	14.7	68.7	12.4	66.5	12.4	60.8	11.1	17.3	< 0.001	0.203	9.2	0.003	0.120
HR max	159.00	16.9	161.0	15.0	156.0	14.8	142.0	13.5	26.1	< 0.001	0.278	45.4	< 0.001	0.401

CTRL = control; SD = standard deviation; CH = carbohydrates; TG = triglycerides; SBP = systolic blood pressure; DBP = diastolic blood pressure; HR = heart rate; max = maximum; F = F statistic; η^2^p = partial eta squared effect sizes. In the model, all differences were significant when *p* < 0.005.

**Table 6 ijerph-19-05857-t006:** Descriptive data (mean ± standard deviation) of quality bone variables. Effect of the intervention (ANOVA) on blood pressure variables.

	CTRL				app									
	Initial		Final		Initial		Final		Effect Time			Effect Time x Period		
	Mean	SD	Mean	SD	Mean	SD	Mean	SD	F	*p*	η^2^_p_	F	*p*	η^2^_p_
BUA (dB/MHz)	133.0	11.4	134.0	12.1	134.0	12.1	135.0	12.2	14.291	<0 .001	0.174	0.335	0.565	0.005
SOS (m/s)	1647.0	35.9	1658.0	38.2	1665.0	38.3	1673.0	38.7	9.033	0.004	0.117	0.230	0.633	0.003
STIFFNESS (A.U)	432.0	15.5	436.0	16.4	439.0	16.5	441.0	16.7	0.416	0.521	0.006	2.314	0.133	0.033

CTRL = control; SD = standard deviation; F = F statistic; η^2^p = partial eta squared effect sizes; BUA: Broadband ultrasound attenuation; SOS: Speed of sound. In the model, all differences were significant when *p* < 0.005.

## Data Availability

The data presented in this study are available on request to the corresponding author. The data are not publicly available due to personal health information.

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
