# Peer review of "Effectiveness of App-Based Intervention to Improve Health Status of Sedentary Middle-Aged Males and Females"

_ijerph, 2022, doi:10.3390/ijerph19105857_

Round 1
Reviewer 1 Report
Thank you for the opportunity to review this manuscript. In general, the idea and rationale of the study is interesting and may have important practical applications for sport scientists and nutritionists working looking to improve self-management of their clients / patients. The paper is well written, and the idea is appealing. However, some general aspects need further addressing, including the writing, building the rationale more effectively, some further details in parts of the methods and results, consideration of addressing some ethical concerns, and wider explanation of findings and practical impact of the outcomes generated in the study.
Title:
Although I don't feel qualified to judge about the English language and style, the sentence “Clinical trial” in the title feels inappropriate. Please consider writing “A Clinical Trial” or just use “Effectiveness of app-based intervention to improve health status”. Another idea would be to add some details about the subjects involved (i.e “…of sedentary middle-aged males and females”, etc.).
Introduction:
The introduction it is long, wordy and not very specific. Although information presented here is correct, most of it is focused on overweight and obesity and the associated problems and the importance of diet and physical activity in relation to these problems. Please consider dedicating most of the introduction to the specific rationale of the study. After a few general sentences on health problems, diet and physical activity it would be nice to read more about the importance of adherence, visualisation of data to improve health and other app-based interventions.
Methods:
After reading the methods section multiple times, I still don’t understand the time-course of the intervention(s) and the (statistical) comparisons made. In the text it says that subjects were both, experimental and control group suggesting a cross-over study design. Furthermore, it is stated that the intervention lasted 6 months and included 3 control periods and 3 intervention periods. Therefore, I understood that control periods and intervention periods were alternated monthly. However, Figure 1 and 2 as well as the statistical section are suggesting a single comparison between interventions. Please clarify.
Additionally, figures 1 and 2 seem to illustrate the exact same thing, only specifying some details differently. Would it be possible to combine these figures into one?
Furthermore, I asked myself if subjects were randomly allocated to one of the intervention groups at the same time or if all of them entered the same intervention all together at once. Figure 2 suggests all of them started the same intervention and then change to the other. However, this would have implications for the interpretation of results and is most probably a limitation. If this is the case, please specify why it was not possible to randomly allocate subjects and carry out both interventions in parallel with half of the sample, respectively.
Finally, and most importantly, the reviewer would like to express ethical concerns about this study. The authors of the study seem to be involved in the development and distribution of the services and application “Nutrición Selecta” as can be seen here: https://www.nutricionselecta.es/equipo
Although this is not a problem per se, funding is clearly stated, and a local ethical committee approved the study the reviewer would like the authors to keep in mind that a special focus on methodological quality is needed. Addressing the above-mentioned points and clearly stating potential limitations is even more important, if economic benefit could be obtained by publishing positive results. Therefore the reviewer would like to refrain from further specific comments before the above mentioned points are clarified.
Author Response
Thank you for your comments. A word document with the answers is attached.

Reviewer 2 Report
Authors have conducted this study with scientific rigor.
The following minor changes must be incorporated into the manuscript:
- The manuscript Title must be redrafted to make it concise and clear to the readers without any ambiguity.
- Stenghts and Limitations of this sytudy must be clearly explained in seperate sections.
- Future directions may be drafted in the manuscript post conclusions section.
Author Response

(The authors gave the same response as above.)

Reviewer 3 Report
Adherence to a nutritional program and physical activity are fundamental aspects of treatment for weight loss and associated problems.
Research in mobile health has yet shown that self-monitoring using a mobile device improves self-management.
One of the problems encountered with wellness apps for self-assessment is that patients tend to proceed without interacting with experienced professionals, blindly trusting the apps.
THESE Scholars propose a reliable approach in this sense, with a connection with the health domain and therefore with experts and with digital health.
Their study provided that: (a) the use of a control group not using the App. (b) All participants had their body composition and cardiovascular variables measured. They also underwent calcaneal densitometry to determine bone quality. (c) a robust statistical approach.
The authors demonstrated that the use of an app mediated by the assistance of dietitians-nutritionists and sports scientists in which a prediction of the evolution, messages of results and advice, among others, has a positive impact on the improvement of health parameters, significant differences were observed in all variables except troponin.
They concluded that combination of healthy habits with the use of the app, give a boost on the benefits, improving the health of the human body.
I have read this article with great interest in an area that I have been following closely for some years.
I found it scientifically well set up, attractive and well written.
With a pure academic spirit I suggest the following changes:
- The article is focused on mobile health, the references cited deal with mhealth. However, in the manuscript you never mention mHealth even if you refer to it. Please include some considerations on mHealth in the introduction and recall it in the manuscript
- Put the keyword mHealth after the abstract.
- The hypotheses are well posed. The aim "In this way, the aim is to provide a solution to a manifest need of population, which would improve the value chain in the health field dedicated to the prevention and treatment of pathologies related to diet and physical exercise, and whose demand is not covered by the national health system. " is too nuanced. I would put a clear purpose
- Methods and results are very well posed and exposed. In the discussion I suggest you include something on the certification aspects of medical devices (see regulation (UE) 2017/745). For example, is it a certified App? In this case give some details. If the App is not certified, however, discuss whether you intend to do it in perspective or if it is not foreseen for some reason (many Wellness Apps are not certified because their intended use does not provide for it)
- In the discussion, I would suggest that you point out in perspective if you intend to do acceptability and or usability tests.
- Still in the discussion, I would discuss the prospects for integration with the health domain and with other digital health solutions.
- Put a description also in English in figure four which has the App menus in Spanish
- Check the resolution of the figures
- Remove from the abstract (1) (2) ..etc.
Thanks to the editors for the opportunity to review this very important study.
Good luck to the authors in continuing their research
Author Response

(The authors gave the same response as above.)

Reviewer 4 Report
The paper presents a study on the effectiveness of an app-based intervention on individuals' with overweight problems. Despite being an interesting read, the manuscript has some flaws, especially in the writing.
I have detailed my comments below:
- Title. My concern is that the title is too generic (“health status”) and does not reflect the specificity of the intervention under investigation (“obesity”). I would suggest the authors to change the title to better reflect the content of the manuscript.
- Introduction. The introduction is clear and concise, and introduce the topic appropriately. However, I would suggest to move some of the articles reported in the discussion section, such as the systematic reviews mention on Line 344, as this would allow the reader to get a better framework of reference for the current presented study.
- Method. My biggest concern with the study is that the author says that they have a control and an experimental group, but this is not the case. What has been measured is instead a before/after intervention change in the same group, but there is no control group. As such, I disagree with the fact that the authors’ can prove it is more effective than a non-app based exposure diet and exercise. In fact, I believe that with the current set of data the authors are not able to exclude the possibility that the greater improvement during the app based period is due to the fact that participants have learned and adapted to the diet and exercise. I believe that the authors should consider either (1) removing the references to the control/experimental group and rewriting the manuscript with a more appropriate terminology or (2) collect a new set of data from participants who are exposed to a double period of non-app based diet and exercise.
- Results. With the previous comment in mind, the result seems strong, and the bonferroni correction confirms the results.
- Discussion. The discussion seems generally appropriate, despite some claims that may not be very scientifically sound (e.g. Line 318 “the difference is statistically superior”) or false/misleading claims (Line 358. “it is a controlled trial, including both control and experimental groups”). As I said before, I believe the manuscript would benefit from a shift of some of the content from this section to the introduction.
Some minor comments:
- Figure 2 is redundant, adds nothing to the manuscript and is not referenced in the text. I would suggest the authors to remove the figure.
- There is no consistency in the way the groups/moments are reported. Sometimes they are referred to as initial and final, sometimes as baseline and finals, and some other times as primary and final. I would suggest the authors to stick to a term to avoid confusion.
- Figure 5. There is a typo “Inicial” instead of “Initial”. Moreover, how is it possible that the same value of 0.12 corresponds to different heights of the bars? I think there is some error on either the y-axis or the labels. Similarly, how can the four bars at 0.11 be at three different levels?
- Sentence on Line 143 is a bit misleading. does that means that all of your participants have cardiovascular problem? Also, a ) is missing.
- Line 208. The sentence is the similar to the one on line 109. I would suggest the authors to remove the the second repetition of the sentence.
Overall, the paper has potential, and i think the topic under investigation can be of interest to a general reader. However, I don’t believe in its current form the paper can be accepted for publication.
Author Response

(The authors gave the same response as above.)

Round 2
Reviewer 1 Report
Thank you for the detailed responses to my initial suggestions and for making relevant changes where suitable - the submission has been greatly improved and reads much strong in this version.
I only have some suggestions to consider further to improve the manuscript further, which are outlined below and are focused on the limitations of the study.
Abstract:
My only suggestion on this front is to consider specifying the intervention. Especially time periods and blinding testers.
Introduction:
Thank you for enhancing the rationale of the study by deleting some paragraphs and adding more information about mobile applications.
Materials and Methods:
Line 113 and Figure 3: Please consider moving this down to 2.5 Study variables.
Lines 161-163: Please add, if possible, software version of Tanita BC-730F used. Please specify measurement preparation (i.e hydration and food intake, time of the day, etc). If no protocol was used, please add to the limitation section.
Lines 172-179: Please add specification (product number or code) of Omron sphygmomanometer, devices for blood analysis and heart rate monitoring. Additionally, please add information regarding stress test protocol. As mentioned before, the reviewer thinks information regarding measurement preparation is crucial here (or should be clearly added as a limitation).
Lines 181-192: Please be more specific by adding references instead of explaining the measurement technique in general. If possible, include information about reliability and validity of the measurement technique.
Limitations: Thank you for adding some lines regarding limitations of the study. However, the reviewer would highly recommend adding more information regarding limitations of the study and / or recommendations for future studies. For more information please refer to Jüni et al. (doi:10.1136/bmj.323.7303.42) CONSORT, SPIRIT checklist, etc.
Especially planned sample size (power), recruitment (responsible persons, expected recruitment rate, monitoring, financial and non-financial incentives, etc.), reliability of the measurements (and therefore observed differences between interventions), blinding of the tester (especially if involved in the company distributing the application), trial design (time-course of the intervention), absence of control group without intervention needs careful attention and must be addressed.
Jüni P, Altman D G, Egger M. Assessing the quality of controlled clinical trials BMJ 2001; 323 :42 doi:10.1136/bmj.323.7303.42
Author Response
Thank you for your comments. Word document with the answers is attached.

Reviewer 4 Report
I am thankful to the authors for having taken my comments into consideration.
I believe that the manuscript has greatly improved as compared to the previous version.
I would suggest the editor to accept it for publication.
Author Response
Thank you for your considerations and comments.